# IMPORTANCE SAMPLING OPTIMIZATION IMPROVES ON-LINE PREFERENCE LEARNING

## ABSTRACT

Training large language models (LLMs) with online sampled data can help off-policy preference optimization approaches like DPO learn better. Recent methods such as Statistical Rejection Sampling Optimization (RSO) have emerged as attractive alternatives to online Reinforcement Learning from Human Feedback (RLHF), offering improvements in stability and scalability. Although RSO has shown promising results by using rejection sampling to obtain preference data from the estimated optimal target policy, it faces computational inefficiencies due to the high rejection rates inherent in its sampling process. To address these limitations, we introduce *Importance Sampling Optimization* (ISO), a novel approach that achieves the benefits of sampling from the optimal policy distribution while significantly improving sample efficiency. ISO employs importance sampling to correct the distribution mismatch between the supervised fine-tuned (SFT) policy and the target optimal policy, enabling efficient use of all generated samples without rejection. Through extensive experiments across diverse tasks and models, we demonstrate that ISO achieves comparable or superior performance to RSO while requiring substantially fewer samples from the SFT policy. Reduces sampling overhead by up to 75% while maintaining or improving win rates against both DPO and RSO baselines. Additionally, we show that ISO naturally extends to other preference optimization methods, providing a general framework for improving sample efficiency in preference learning.

## 1 INTRODUCTION

Recent advances in Large Language Models (LLMs) have pushed the boundaries of downstream performance, enabling unprecedented capabilities across a wide variety of tasks, including reasoning, summarization, and dialogue systems (Achiam et al., 2023; Touvron et al., 2023; Google, 2023; Anthropic, 2024; Guo et al., 2025). A key factor in their success has been alignment, the process of steering model behavior to be more helpful and harmless. Reinforcement Learning from Human Feedback (RLHF) stands out as a canonical approach for aligning LLMs with human preferences, enjoying widespread adoption for improving supervised fine-tuned (SFT) models (Ouyang et al., 2022; Stiennon et al., 2020; Gao et al., 2024). However, the classical online RLHF framework is computationally expensive and involves intricate training pipelines, including reward modeling and policy training (Yuan et al., 2023; Dong et al., 2024; Liu et al., 2023). To circumvent these complexities, simpler offline methods have emerged, such as Direct Preference Optimization (DPO) (Rafailov et al., 2024) and Sequence Likelihood Calibration (SLiC) (Zhao et al., 2023), achieving strong performance by directly optimizing a policy on a static, pre-collected preference dataset without explicit RL training.

While offline methods like DPO are resource-efficient, they rely on a fixed dataset, which may not adequately represent the distribution of the model being trained. A key innovation to address this is the use of online alignment, where synthetic preference data is generated directly from the policy as it undergoes training (Abdin et al., 2024; Setlur et al., 2024). This online approach ensures the training distribution remains close to the current policy, mitigating distributional discrepancies and leading to improved generation quality and training stability. Recent studies have consistently shown that such online methods can considerably outperform purely offline approaches, establishing them as a more potent paradigm for LLM alignment(Tang et al., 2024a; Tajwar et al., 2024).

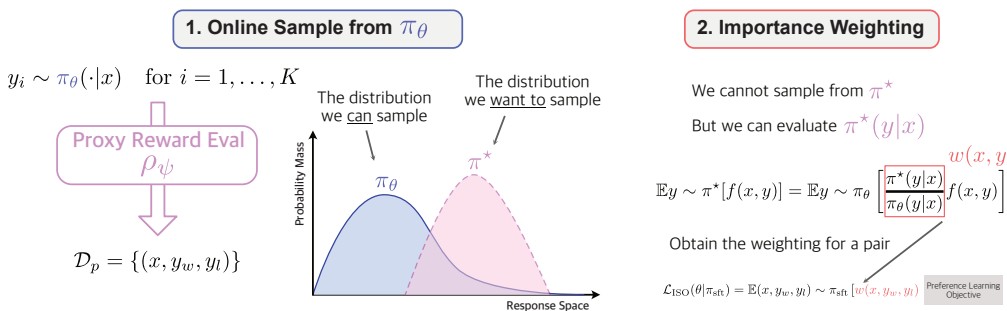

Figure 1: An overview of Importance Sampling Optimization (ISO). We sample preference pairs from the current policy $\pi_\theta$, which deviates from the optimal target policy $\pi^*$. Then use importance weighting to adjust the DPO loss for each pair, thereby correcting the distributional mismatch and improving optimization.

However, applying online data to existing frameworks introduces a new challenge. Methods like DPO are designed to learn an optimal policy, $\pi^*$, but the online data is necessarily sampled from the current, non-optimal policy, $\pi_\theta$. This creates a fundamental distribution mismatch. To bridge this gap, RSO(Liu et al., 2023) was recently proposed. RSO attempts to correct this mismatch by using a reward model to implement rejection sampling. It generates candidate responses from the current policy $\pi_\theta$ and then probabilistically accepts or rejects them to create a new dataset that more closely approximates samples from the target optimal policy, $\pi^*$.

Although RSO demonstrates the promise of correcting the online data distribution, its reliance on rejection sampling is a significant bottleneck. The process can be highly inefficient, discarding a large fraction of the generated samples, especially when the current policy is far from the optimal one. This wastefulness translates directly into increased computational cost and slower training cycles. This inefficiency highlights the need for an approach that can retain the benefits of online distribution correction without the high cost of rejection sampling. A powerful and well-established technique for correcting distribution shifts without discarding data is importance sampling.

Inspired by this, we introduce Importance Sampling Optimization (ISO), a novel online alignment framework that replaces inefficient rejection sampling with principled importance sampling as shown in Figure 1. ISO corrects the distributional mismatch between the current sampling policy ($\pi_\theta$) and the target optimal policy ($\pi^*$) by re-weighting the loss for each preference pair. By assigning an "importance weight" to each sample, ISO can effectively leverage all generated data, eliminating the computational waste inherent in RSO while still optimizing towards the true underlying preference distribution. Our contributions are:

1. We propose ISO, a new online alignment algorithm that uses importance sampling to create a more sample-efficient and computationally-efficient alternative to RSO.

2. We derive the specific formulation for the importance weights, allowing for a seamless integration into the DPO loss function to correct for the online distribution shift. We also introduce a signed margin score to increase the weight of more informative preference pairs, which mitigates the issue that sampled response pairs tend to have a low margin.

3. Through extensive experiments, we demonstrate that ISO consistently matches or exceeds the performance of existing alignment methods, including RSO and offline DPO, while requiring significantly less computational overhead.

## 2 BACKGROUND

Our work builds upon existing methods for aligning LLMs with human preferences. This section first discusses the reward model training using the human preference datasets in Section 2.1. We then discuss the preference optimization algorithms in Section 2.2, and the online improvement of the

preference learning in Section 2.3. This future motivates RSO to refine the sample distribution, which is discussed in Section 2.4.

## 2.1 REWARD MODELING

**Pairwise Reward Model** In our pipeline, we use a pairwise reward model to serve as a proxy for human preferences. We train a sequence-to-sequence model, $\rho_\psi$ (specifically, a Flan-T5-XXL), to predict which of two responses is preferred for a given prompt $x$. The model takes a formatted input containing the prompt and both responses $(y_1, y_2)$ and outputs logits for "A" (preferring $y_1$) and "B" (preferring $y_2$). From these logits, we can estimate the preference probability:

$$\hat{P}(y_1 \succ y_2|x) = \sigma(\text{logitA} - \text{logitB}), \tag{1}$$

where $\sigma$ is the sigmoid function. To mitigate potential positional bias, the training data is augmented by swapping the positions of $y_1$ and $y_2$.

While this model is inherently pairwise, we can induce a pointwise reward score, $r_\psi(x, y)$, which is necessary for methods like RSO. This is done by comparing a given response $y$ against a fixed baseline response $y_b$, whose reward is assumed to be zero. The score is then calculated as:

$$r_\psi(x, y) = \text{logitA} - \text{logitB} \quad \text{where } y_1 = y, y_2 = y_b.$$

In practice, we use a random response from the SFT policy as the baseline $y_b$.

## 2.2 PREFERENCE OPTIMIZATION FOR LLM ALIGNMENT

The standard paradigm for aligning LLMs with human values has been RLHF. The RLHF process typically involves three main steps: 1) collecting a dataset of human preferences between pairs of model responses, 2) training a separate reward model (RM) to predict which responses humans would prefer, and 3) fine-tuning the LLM policy using reinforcement learning (e.g., PPO) to maximize the score from the RM(Ouyang et al., 2022). While effective, the RLHF pipeline is known to be computationally expensive and often suffers from training instability

DPO was introduced as a more stable and efficient alternative that bypasses the need for an explicit reward model and the complexities of reinforcement learning. We use the DPO framework to align the policy $\pi_\theta$ with the generated preference data $\mathcal{D}_p$(Rafailov et al., 2024). DPO established a direct mapping from preference probabilities to an optimal policy, reframing the alignment task as a simple classification problem on preference pairs. The standard DPO loss is derived from the Bradley-Terry model and is formulated as a negative log-likelihood loss:

$$\mathcal{L}_{\text{DPO}}(\pi_\theta|\pi_{\text{sft}}) = \mathbb{E}_{(x,y_w,y_l) \sim \mathcal{D}_p}\left[-\log \sigma\left(\beta \log \frac{\pi_\theta(y_w|x)}{\pi_{\text{sft}}(y_w|x)} - \beta \log \frac{\pi_\theta(y_l|x)}{\pi_{\text{sft}}(y_l|x)}\right)\right], \tag{2}$$

where $\sigma$ is the sigmoid function. In addition to this sigmoid loss, recent work like SLiC (Zhao et al., 2023) also uses a normalized hinge loss variant:

$$\mathcal{L}_{\text{DPO}}(\pi_\theta|\pi_{\text{sft}}) = \mathbb{E}_{(x,y_w,y_l) \sim \mathcal{D}_p}\left[\max\left(0, \delta - \beta \log \frac{\pi_\theta(y_w|x)}{\pi_{\text{sft}}(y_w|x)} + \beta \log \frac{\pi_\theta(y_l|x)}{\pi_{\text{sft}}(y_l|x)}\right)\right], \tag{3}$$

where $\beta$ is the DPO temperature and $\delta$ is the hinge margin, typically set to 1. These objectives effectively increase the relative probability of the preferred response $y_w$ over the dispreferred response $y_l$. To make training computationally feasible, Low-Rank Adaptation (LoRA) (Hu et al., 2022) can be employed, which drastically reduces the number of trainable parameters by updating low-rank adapter matrices instead of the full model weights.

## 2.3 ONLINE PREFERENCE LEARNING

While DPO simplifies the alignment process, the standard offline approach relies on a static, pre-collected dataset of preferences. This creates a fundamental challenge known as **distribution shift**(Li et al., 2025; Guo et al., 2024). As the policy $\pi_\theta$ is updated during training, its output distribution

diverges from the one that generated the static dataset, making the training data increasingly "off-policy" and less relevant to the model's current capabilities and failure modes. This mismatch can lead to suboptimal performance and training instability.

To address these limitations, **online preference learning** frameworks have been proposed(Xiong et al., 2023; Guo et al., 2024; Dong et al., 2024). These methods integrate data generation into the training loop, ensuring that the model is continuously learning from feedback on its own most recent outputs. Our work builds upon a multi-stage pipeline for aligning LLMs using such online preference data.

**Supervised Fine-tuning (SFT)** The alignment process begins with a pre-trained LLM, which is fine-tuned on a high-quality, instruction-following dataset $\mathcal{D}_{\text{sft}} = \{(x, y_{\text{chosen}})\}$. This produces the SFT model, $\pi_{\text{sft}}$, by minimizing the standard negative log-likelihood loss:

$$\mathcal{L}_{\text{sft}}(\theta) = -\mathbb{E}_{(x, y_{\text{chosen}}) \sim \mathcal{D}_{\text{sft}}} \left[ \log \pi_\theta(y_{\text{chosen}}|x) \right]. \tag{4}$$

The resulting $\pi_{\text{sft}}$ serves as both the initial policy for alignment and as a reference policy during preference optimization to prevent divergence from the learned distribution.

**Online Preference Data Generation** To mitigate the distribution mismatch inherent in static, offline datasets, online methods generate preference data iteratively. In each iteration, a set of $K$ candidate responses $\{y_1, \ldots, y_K\}$ is sampled from the current policy $\pi_\theta$ for a given prompt $x$. These responses are then ranked using the pairwise reward model $\rho_\psi$ to create a preference dataset $\mathcal{D}_p = \{(x, y_w, y_l)\}$, where $y_w$ is preferred over $y_l$. We adopt the "first-round-rank" strategy from Liu et al. (2023), where all candidate pairs are ranked to form the final dataset.

## 2.4 STATISTICAL REJECTION SAMPLING OPTIMIZATION (RSO)

Statistical Rejection Sampling Optimization (RSO) (Liu et al., 2023) refines the online data generation process to better approximate the optimal policy $\pi^\star$. Instead of using all sampled candidates, RSO uses rejection sampling to filter them. It accepts a candidate $y_i$ sampled from $\pi_\theta$ with a probability proportional to $\exp(r_\psi(x, y_i)/\beta)$, where $r_\psi$ is the pointwise reward and $\beta$ is a temperature parameter. This filtering step aims to create a dataset that more closely follows the distribution of the target optimal policy. While effective, this process can be highly sample-inefficient, as many candidates may be rejected and discarded.

## 3 METHOD

In this section, we present ISO for efficient online alignment of language models. We first formulate the distribution mismatch problem when sampling from the current policy rather than the optimal policy (Section 3.1). We then derive importance weights to correct this mismatch, introducing a signed margin score to prioritize informative preference pairs (Section 3.2). Next, we integrate these weights into the preference optimization loss (Section 3.3) and describe ISO's implementation as a plug-and-play module for existing pipelines (Section 3.4). We include the detailed derivation in the Appendix A.1.

## 3.1 STATISTICAL FORMULATION

While RSO (Liu et al., 2023) uses rejection sampling to draw samples from this optimal policy, this approach leads to high rejection rates and computational inefficiency. Instead, we propose to correct for the distribution mismatch between $\pi_{\text{sft}}$ and $\pi^\star$ using importance sampling.

The core challenge lies in estimating expectations under $\pi^\star$ when we can only sample from $\pi_{\text{sft}}$. For any function $f(x, y)$, this expectation can be written as:

$$\mathbb{E}y \sim \pi^\star[f(x, y)] = \mathbb{E}y \sim \pi_{\text{sft}} \left[ w(x, y) f(x, y) \right] \tag{5}$$

where $w(x, y) = \frac{\pi^\star(y|x)}{\pi_{\text{sft}}(y|x)}$ is the importance weight.

## 3.2 Importance Sampling for Preference Optimization

The derivation of importance weights begins with the optimal policy for a single response, which can be expressed as:

$$w(x, y) = \frac{1}{Z(x)} \exp\left(\frac{1}{\gamma} r(x, y)\right) \tag{6}$$

where $r(x, y)$ is the reward for response $y$ to prompt $x$, and $Z(x)$ is the partition function over all possible responses. $\gamma$ is the ISO temperature hyperparameter that controls the importance weight distribution. It should be decided by how much we trust the reward model. The more accurate and robust the reward model is, the smaller the value of $\gamma$ should be. We show

For preference pairs $(y_w, y_l)$ consisting of a winning and losing response, we must consider their joint distribution. Assuming the responses are sampled independently conditioned on the prompt $x$, the unnormalized pair-wise importance weight is initially computed based on the sum of their rewards:

$$\tilde{w}_{base}(x, y_w, y_l) = \exp\left(\frac{1}{\gamma}(r(x, y_w) + r(x, y_l))\right) \tag{7}$$

However, not all preference pairs are equally informative. A pair where the reward margin $(r(x, y_w) - r(x, y_l))$ is large and positive is more consistent with the learned reward function than a pair where the margin is small or negative. To incorporate this, we introduce a modulating factor based on the Bradley-Terry model, which represents the probability of $y_w$ being preferred over $y_l$:

$$P(y_w \succ y_l | x) = \sigma(r(x, y_w) - r(x, y_l))$$

where $\sigma(\cdot)$ is the sigmoid function. By centering this probability around 0.5, we design a **signed margin score** $(\sigma(r(x, y_w) - r(x, y_l)) - 0.5)$ which is positive when $r_w > r_l$ and approaches zero when the rewards are nearly equal. This score down-weights pairs with a low reward margin, reducing their impact on the overall objective. Our final unnormalized weight, $\tilde{w}$, is the product of the base weight and this margin score:

$$\tilde{w}(x, y_w, y_l) = \tilde{w}_{base}(x, y_w, y_l) \cdot (\sigma(r(x, y_w) - r(x, y_l)) - 0.5) \tag{8}$$

To handle the partition function and ensure numerical stability, we normalize these weights within each prompt's set of preference pairs:

$$w(x, y_w, y_l) = \frac{\tilde{w}(x, y_w, y_l)}{\sum_{(y'_w, y'_l)} \tilde{w}(x, y'_w, y'_l)} \tag{9}$$

This normalization ensures that the weights for all preference pairs originating from the same prompt sum to one, effectively removing the dependency on the partition function $Z(x)$.

## 3.3 Loss Function with Importance Weights

We modify the sigmoid loss function to incorporate importance weights:

$$\mathcal{L}_{\text{ISO}}(\theta | \pi_{\text{sft}}) = \mathbb{E}_{(x, y_w, y_l) \sim \pi_{\text{sft}}} \left[ -w(x, y_w, y_l) \log \sigma \left( \beta \log \frac{\pi_\theta(y_w | x)}{\pi_{\text{sft}}(y_w | x)} - \beta \log \frac{\pi_\theta(y_l | x)}{\pi_{\text{sft}}(y_l | x)} \right) \right] \tag{10}$$

where the normalized importance weights $w(x, y_w, y_l)$ correct for the distribution mismatch between $\pi_{\text{sft}}$ and $\pi^\star$, and $\beta$ controls the margin scaling. We can also incorporate the importance weight into the hinge-norm loss following the same pattern.

## 3.4 Integration into Preference Learning Pipelines

ISO is designed as a plug-and-play module that seamlessly integrates into existing online preference optimization pipelines. The method only requires computing importance weights from reward values and applying them as sample-level multipliers during policy updates, making it compatible with any DPO-based framework without architectural modifications. Algorithm 1 presents the complete ISO training procedure, where highlighted lines show the key ISO-specific implementations: importance weight computation (line 8) and per-prompt normalization (line 12).

---

**Algorithm 1** Importance Sampling Optimization (ISO)

---

**Require:** Initial policy $\pi_{\theta_0} = \pi_{\text{sft}}$, reward model $r_\phi$, prompts $\mathcal{X}$, iterations $T$
**Ensure:** Aligned policy $\pi_{\theta_T}$
 1: **for** $t = 0$ to $T - 1$ **do**
 2:     $\mathcal{D}_t \leftarrow \emptyset$                                            ▷ Initialize online dataset
 3:     **for** each prompt $x \in \mathcal{X}$ **do**
 4:         Sample $K$ responses: $\{y^k\}_{k=1}^K \sim \pi_{\theta_t}(\cdot|x)$
 5:         Compute rewards: $\{r^k\}_{k=1}^K$ where $r^k = r_\phi(x, y^k)$
 6:         Construct pairs: $\mathcal{P}_x = \{(y^i, y^j) : r^i > r^j\}$
 7:         **for** each pair $(y_w, y_l) \in \mathcal{P}_x$ **do**
 8:             $\tilde{w} \leftarrow \exp\left(\frac{1}{\gamma}(r_w + r_l)\right) \cdot (\sigma(r_w - r_l) - 0.5)$
 9:             $\mathcal{D}_t \leftarrow \mathcal{D}_t \cup \{(x, y_w, y_l, \tilde{w})\}$
10:         **end for**
11:     **end for**
12:     Normalize weights: $w_i \leftarrow \tilde{w}_i / \sum_j \tilde{w}_j$ for all samples with same $x$
13:     Update policy: $\theta_{t+1} \leftarrow \arg\min_\theta \mathcal{L}_{\text{ISO}}(\theta; \mathcal{D}_t, \{w_i\})$
14: **end for**
15: **return** $\pi_{\theta_T}$

---

**Compatibility.** ISO's importance weighting mechanism is loss-agnostic and can be applied to various preference optimization objectives beyond DPO, including IPO, SLiC, and other alignment methods. The only requirement is access to reward scores, which are typically already computed in online preference learning pipelines. This out-of-the-box design allows ISO to enhance sample efficiency without modifying the underlying optimization algorithm or model architecture, users simply replace the standard loss with the weighted version $w_i \cdot \mathcal{L}(\cdot)$.

# 4 EXPERIMENTS

We conduct comprehensive experiments to evaluate ISO's performance across multiple models and datasets, comparing it against other methods. Our evaluation focuses on three key aspects: response quality, sample efficiency, and robustness across different training configurations.

## 4.1 EXPERIMENT SETUP

**Golden Reward Model** To provide an unbiased and robust assessment of the different alignment methods, we employ a powerful, held-out reward model as a "golden" evaluator. We fine-tune a **Gemma-2-27B** model, initialized from the `Skywork/Skywork-Reward-Gemma-2-27B-v0.2` checkpoint, on our target preference dataset. This model is trained as a **pointwise scorer**, $r_{\text{eval}}(x, y)$, using a standard negative log-likelihood loss based on the Bradley-Terry model. This objective trains the model to predict a scalar score for a given response such that the difference in scores between two responses accurately reflects the human preference probability. Crucially, this evaluation model is **completely independent** of the training process; it is never used to generate preference labels or provide rewards during the alignment of any of the policies, including our proposed ISO method and all baselines. This separation ensures a fair and objective comparison, where performance is measured by a consistent and powerful judge.

**Models and Datasets** We evaluate ISO on three model families: GPT-2 (Radford et al., 2019), Gemma-2 (Team et al., 2024), and Qwen2.5 (Qwen et al., 2024). Our experiments utilize two standard alignment benchmarks: Reddit TL;DR (Stiennon et al., 2020) and UltraFeedback (Cui et al., 2023), both widely adopted in the alignment literature (Yuan et al., 2024; Jian et al., 2025; Liu et al., 2024; Dong et al., 2024). We maintain the original train-test splits, with all results reported on held-out test sets. Both reward model training and preference optimization use only training data.

Table 1: Compare different methods on the Reddit TL;DR and Ultrafeedback dataset. The number of samples used in the preference learning is 16. The RSO is required to rollout 64 samples from the SFT policy. The Direct and ISO are required to rollout 16 samples. Win rates against responses generated from the corresponding SFT policy are reported, evaluated using the proxy reward model. We show both the average win rate evaluated using the proxy and the golden reward in the last two columns. We find that our ISO method consistently achieves a higher win rate compared to the baseline methods on both criteria of proxy reward and golden reward. Full golden reward win rates for individual models are provided in Table 3 in the appendix.

| Iteration | Method | Gemma2 | | GPT2 | | Qwen2.5 | | Average Win Rate | |
|---|---|---|---|---|---|---|---|---|---|
| | | 2B | 9B | Large | Medium | 1.5B | 3B | **Proxy** | Golden |
| | | | | | Reddit TL;DR | | | | |
| | **Direct** | 86.99 | 92.54 | 55.25 | 48.86 | 73.99 | 79.86 | 72.92 | 57.51 |
| | **RSO** | 91.08 | 88.63 | 53.27 | 49.44 | 78.00 | 82.60 | 73.84 | 57.20 |
| # 1 | **ISO** (ours) | 93.28 | 94.76 | 54.74 | 48.81 | 82.40 | 87.94 | **76.99** | **59.29** |
| | | | | | Ultrafeedback | | | | |
| | **Direct** | 80.77 | 85.07 | 52.49 | 49.72 | 71.62 | 79.74 | 69.90 | 54.96 |
| | **RSO** | 80.14 | 84.84 | 50.73 | 49.02 | 67.55 | 78.08 | 68.39 | 52.67 |
| | **ISO** (ours) | 85.62 | 90.52 | 52.09 | 50.03 | 77.50 | 84.74 | **73.42** | **56.01** |
| | | | | | Reddit TL;DR | | | | |
| | **Direct** | 94.01 | 94.84 | 60.01 | 49.71 | 84.60 | 91.04 | 79.04 | 58.35 |
| | **RSO** | 93.87 | 91.86 | 57.54 | 50.01 | 88.61 | 91.80 | 78.95 | 57.49 |
| # 2 | **ISO** (ours) | 95.16 | 96.28 | 59.49 | 49.95 | 91.50 | 94.66 | **81.17** | **59.15** |
| | | | | | Ultrafeedback | | | | |
| | **Direct** | 89.47 | 89.74 | 55.25 | 49.80 | 83.36 | 90.65 | 76.38 | 52.19 |
| | **RSO** | 90.25 | 87.03 | 52.74 | 49.87 | 82.05 | 86.80 | 74.79 | 51.07 |
| | **ISO** (ours) | 95.05 | 92.33 | 55.56 | 49.17 | 86.48 | 93.77 | **78.73** | **54.38** |

**Evaluation Metrics** Following Liu et al. (2023), we report win rates against SFT-generated responses. For each test prompt, we generate responses from both $\pi_{sft}$ and the current policy $\pi_\theta$, then evaluate them using both proxy and golden reward models. The win rate represents the proportion of $\pi_\theta$ responses that achieve higher rewards than their $\pi_{sft}$ counterparts.

We include the detailed training configuration in the Appendix A.3.

## 4.2 RESULTS AND ANALYSIS

**ISO vs. Other Approaches** Table 1 compares three sampling approaches at fixed preference-learning budgets: Direct (uniform weighting of on-policy samples), RSO (rejection sampling from optimal distribution), and ISO (importance sampling from optimal distribution). ISO achieves the strongest average performance under both proxy and golden reward evaluation, while requiring only **25% of RSO's computational budget** for sample generation. The Direct method, equivalent to ISO with $\gamma \to \infty$ and no margin scoring, uniformly weights all samples. While all methods optimize toward the proxy reward (resulting in higher proxy win rates), ISO demonstrates superior robustness by maintaining strong golden reward performance compared to both RSO and Direct approaches. This indicates ISO's ability to balance proxy optimization with genuine response quality.

**Scaling with Sample Size** Table 2 demonstrates ISO's performance across different sampling budgets. The **# of samples** column indicates responses generated from $\pi_\theta$ (or $\pi_{sft}$ in round one). Proxy win rates improve monotonically with increased sampling, validating our theoretical prediction that larger sample sizes reduce importance sampling variance, thereby enhancing preference learning quality. In practice, we find that 16 rollouts provide an excellent efficiency-quality trade-off for resource-constrained settings. This scaling behavior demonstrates ISO's flexibility in adapting to different computational budgets.

Table 2: Compare different sampling sizes on the Reddit TL;DR and Ultrafeedback dataset using ISO. The reported metrics are the same as Table 1. We choose the sample size to be 16, which significantly reduces the rollout and reward evaluation computation cost and maintains a reasonable generalization performance on the golden reward. Full golden reward win rates for individual models are provided in Table 4 in the appendix. Results for iteration 2 are provided in Table 5 in the appendix.

| Number of Samples | Gemma2 | | GPT2 | | Qwen2.5 | | Average Win Rate | |
| --- | --- | --- | --- | --- | --- | --- | --- | --- |
| | 2B | 9B | Large | Medium | 1.5B | 3B | **Proxy** | Golden |
| Reddit TL;DR | | | | | | | | |
| **8** | 88.73 | 93.77 | 52.43 | 48.67 | 72.07 | 78.08 | 72.29 | 58.38 |
| **16** | 93.28 | 94.76 | 54.74 | 48.81 | 82.40 | 87.94 | 76.99 | 59.29 |
| **32** | 89.68 | 89.69 | 60.69 | 49.90 | 82.10 | 93.13 | 77.53 | **60.60** |
| **64** | 85.90 | 86.53 | 69.86 | 50.83 | 86.42 | 89.68 | **78.20** | 58.67 |
| Ultrafeedback | | | | | | | | |
| **8** | 79.54 | 82.25 | 52.04 | 48.62 | 71.02 | 77.65 | 68.52 | 54.77 |
| **16** | 85.62 | 90.52 | 52.09 | 50.03 | 77.50 | 84.74 | 73.42 | **56.01** |
| **32** | 89.47 | 87.25 | 53.34 | 50.20 | 82.48 | 88.96 | 75.28 | 55.70 |
| **64** | 91.68 | 86.65 | 57.82 | 50.40 | 86.58 | 87.18 | **76.72** | 54.37 |

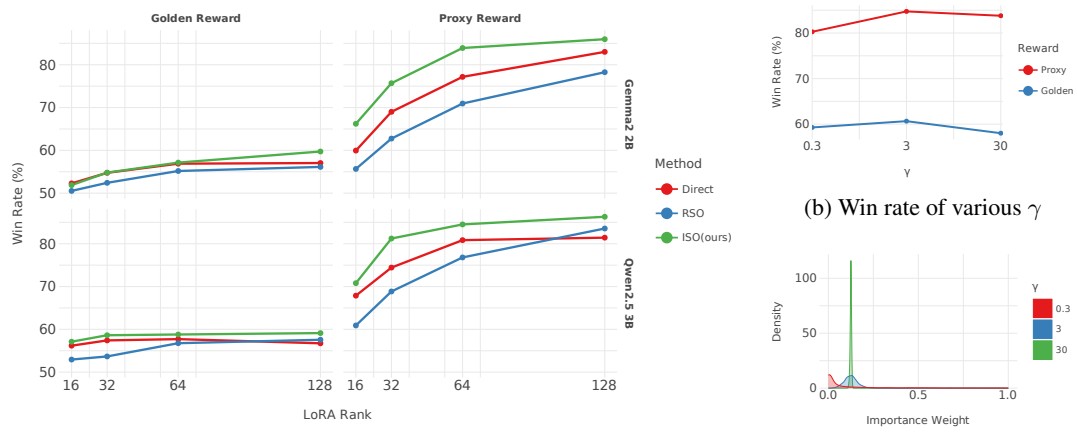

(a) Win rate of various LoRA rank across different methods

(b) Win rate of various $\gamma$

(c) $w_{base}$ distribution of various $\gamma$

Figure 2: Performance analysis of ISO. (a) Win rates versus LoRA rank for different methods (iteration 1, Ultrafeedback dataset). ISO consistently outperforms baselines across all LoRA ranks, while RSO shows degradation at lower ranks. (b) Effect of the ISO temperature parameter $\gamma$ on win rates on the Ultrafeedback dataset with Qwen2.5 3B policy model. Smaller $\gamma$ values (indicating higher trust in the reward model) yield better performance with the proxy reward model, while the golden reward model shows more stability across $\gamma$ values. (c) Distribution of importance weights $w_{base}$ across different $\gamma$ values, showing how smaller $\gamma$ produces more concentrated weights while larger $\gamma$ leads to more uniform weighting. The $w_{base}$ before multiplying the signed margin score is shown for better visualization.

**LoRA Integration**   Figure 2a shows ISO's performance with parameter-efficient fine-tuning via LoRA (Hu et al., 2022). We increase the learning rate to 5e-5 (standard practice for LoRA) and reduce $\beta$ to 0.05 to account for LoRA's structural regularization. ISO consistently outperforms baselines across all LoRA ranks, with performance improving monotonically as rank increases.

**Effect of $\gamma$ in ISO**   Figure 2b examines the effect of $\gamma$ used in Equation 7. We test three different $\gamma$ settings and find that $\gamma = 3$ achieves the best performance on both proxy and golden reward metrics. This parameter controls the distribution of importance weights, with lower values indicating higher trust in the reward model. Figure 2c visualizes the corresponding importance weight distributions for

these three $\gamma$ settings. High $\gamma$ produces near-uniform weights across samples, making the distribution closer to the original on-policy distribution (equivalent to the Direct setting). Low $\gamma$ downweights most samples, creating a density peak near 0, with only a few samples receiving very high weights near 1. The optimal setting of $\gamma = 3$ strikes a balance between these extremes.

## 5 RELATED WORK

**RLHF and preference optimization** Reinforcement Learning from Human Feedback (RLHF) has emerged as a canonical approach for aligning Large Language Models (LLMs) with human preferences (Ouyang et al., 2022; Stiennon et al., 2020; Gao et al., 2024). The classical RLHF framework, initially developed by Christiano et al. (2017) and later popularized in models like InstructGPT (Ouyang et al., 2022), Claude (Bai et al., 2022), and LLaMA-2 (Touvron et al., 2023), typically involves three phases: supervised fine-tuning, reward model training, and policy optimization using algorithms like Proximal Policy Optimization (PPO). Despite its effectiveness, RLHF with PPO presents significant challenges, including training instability (Choshen et al., 2019), sensitivity to implementation details (Engstrom et al., 2020), and high computational requirements (Yuan et al., 2023). To address these limitations, offline preference optimization methods such as Sequence Likelihood Calibration (SLiC) (Zhao et al., 2023) and Direct Preference Optimization (DPO) (Rafailov et al., 2024) have been proposed, offering improved stability and efficiency by directly optimizing language models on preference data without requiring separate reward modeling. Further developments include IPO (Azar et al., 2023), KTO (Ethayarajh et al., 2024), ARM (Pang et al., 2024), and GPO (Tang et al., 2024b). While efficient, these offline methods typically operate on preference datasets collected from other models, potentially leading to distribution mismatch. Recent research demonstrates that online iterative variants of these algorithms significantly outperform their offline counterparts (Xiong et al., 2023; Guo et al., 2024; Xu et al., 2023; Tajwar et al., 2024; Dong et al., 2024), with Statistical Rejection Sampling Optimization (RSO) (Liu et al., 2023) addressing this limitation by implementing rejection sampling to generate preference pairs from an approximation of the optimal policy distribution.

**Importance Sampling** Importance sampling is a fundamental statistical technique for estimating properties of a target distribution using samples from a different behavioral distribution (Levine et al., 2020). In the context of reinforcement learning, importance sampling has been extensively used for off-policy evaluation (Thomas et al., 2015; Thomas & Brunskill, 2016), where the goal is to estimate the performance of a target policy using data collected from a behavioral policy by correcting the distribution with importance weights $w_{P/Q}(x) = p(x)/q(x)$ (Levine et al., 2020). These importance weights serve as a correction mechanism that enables accurate estimation despite the distribution mismatch between the sampling and target distributions. Jiang et al. (2025) uses importance weighting to identify and filter out self-generated samples with high distribution shift extent in language model self-improvement. PILAF (Feng et al., 2025) sample response pair that aligns preference learning with maximizing the underlying reward. DFT (Wu et al., 2025) rewriting SFT gradient as policy gradient via importance sampling. IW-DPO (Lodkaew et al., 2025) uses importance weighting to address deployment distribution shifts between training and test environments by reweighting a fixed offline preference dataset based on a small validation set from the target distribution. Our approach differs from previous works by addressing the distribution mismatch between the current policy and the theoretical optimal policy defined by the reward model, enabling more efficient utilization of all online generated samples.

## 6 CONCLUSION

The proposed ISO method refines the distribution of online sampled data from the SFT policy. It is designed to fit the MLE of DPO loss with the data distribution of the optimal policy. It offers better sample efficiency than other alternatives like rejection sampling. We show its effectiveness across different preference learning datasets and different scales of models. Future work can utilize ISO's efficient distribution correction mechanism in non-preference online RL methods on tasks such as reasoning, coding, and multi-turn interaction, where the ability to leverage all generated samples while accurately targeting the optimal policy could yield significant improvements.

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
