# OpenReview forum: "Importance Sampling Optimization Improves Online Preference Learning"
_ICLR.cc/2026/Conference — Submitted to ICLR 2026_

### Official Review · Reviewer_J7oq · 2025-10-26

**Soundness:** 3
**Presentation:** 3
**Contribution:** 2
**Rating:** 4
**Confidence:** 4

**Summary:**

The paper develops the importance sampling optimization (ISO) approach, to correct the mismatch between the SFT policy and the target optimal policy. ISO finds a way to use all generated samples without rejection, thus differs from the existing RSO (rejection sampling optimization) and DPO approaches, and strikes a balance between sampling efficiency and improved performance in preference learning.

**Strengths:**

Importance sampling is a well-established technique in stochastic optimization, including RL. Thus, ISO is built upon a solid theoretical foundation. The paper is well motivated (starting from Fig 1) and clearly written. The key is the pairwise importance weight in (7), modulated by the signed margin score in (8), which then (after normalization) goes into the loss function in (9), and then integrated into the preference learning pipeline. All these are clearly and logically spelled out in \S 3.

**Weaknesses:**

Cannot help feeling the paper's contribution falls a bit thin on technical novelty, given the well established status of importance sampling.

**Questions:**

At the end of \S4, there’s some description of the effect of \gamma in ISO, via (7). Wonder what’s choice of \beta in (10) in this case?

---

> ### Author Response · Authors · 2025-11-21
> **Response to Reviewer J7oq**
>
> We thank the reviewer for their evaluation. We address the concerns regarding novelty and the hyperparameter details below.
>
> > Cannot help feeling the paper's contribution falls a bit thin on technical novelty, given the well established status of importance sampling.
>
> We acknowledge that while Importance Sampling (IS) itself is a classical statistical technique, our contribution lies in identifying how to rigorously apply it to solve a critical inefficiency in modern **Online Preference Optimization**, specifically as a superior alternative to Rejection Sampling Optimization (RSO).
>
> We argue that the novelty should be viewed through three lenses:
> 1.  **Theoretical Bridge:** We provide the derivation that links the *implicit* optimal policy (defined by the reward model and SFT reference) to the online sampling distribution. This allows us to replace the computationally wasteful rejection sampling process of RSO (which discards ~75% of generated data) with a principled re-weighting scheme that utilizes **100%** of the samples.
> 2.  **Algorithmic Adaptation (Signed Margin Score):** Standard IS can be unstable in pairwise settings. We introduce a novel **Signed Margin Score** (Eq. 8) specifically designed for preference optimization. As shown in our ablation studies (Table 1 in the General Response), this component is critical for performance, effectively modulating the IS weights to prioritize informative preference pairs over noisy ones.
> 3.  **Practical Impact:** We show that this mathematically simple change leads to substantial efficiency gains (4x sample efficiency compared to RSO) without sacrificing performance. We believe applying a classic principle to remove a major bottleneck in LLM alignment constitutes a significant practical contribution.
>
> > At the end of \S4, there’s some description of the effect of $\gamma$ in ISO, via (7). Wonder what’s choice of $\beta$ in (10) in this case?
>
> For the experiments discussed in Section 4 (specifically the non-LoRA settings), we maintained the standard DPO setting of **$\beta = 0.1$** in Equation 10.
>
> We kept $\beta$ fixed to ensure a fair, controlled comparison with the baselines (Direct DPO and RSO), which also use $\beta=0.1$.

---

### Official Review · Reviewer_YRTE · 2025-10-27

**Soundness:** 2
**Presentation:** 3
**Contribution:** 2
**Rating:** 4
**Confidence:** 4

**Summary:**

This paper targets the sample inefficiency of online preference learning methods like Statistical Rejection Sampling Optimization (RSO), which uses costly rejection sampling to align the sampling distribution ($\pi_\theta$) with the target optimal policy ($\pi^*$). The authors propose Importance Sampling Optimization (ISO), replacing rejection sampling with importance sampling to correct the distribution mismatch. ISO computes reward-based importance weights, allowing a DPO-style loss to utilize all generated samples efficiently. A heuristic "signed margin score" is added to potentially upweight informative pairs.

**Strengths:**

1. This paper leverages a importance sampling to solve the distribution mismatch problem and tackles the significant sample inefficiency and computational cost associated with RSO.
2. This paper demonstrates substantial reductions in the required number of sampled responses compared to RSO while maintaining or improving alignment performance. This is a major practical advantage.
3. Experiments are conducted across multiple model families, sizes, and standard alignment datasets, lending credibility to the results. The use of both proxy and independent golden reward models for evaluation adds robustness.

**Weaknesses:**

1. There is a critical misalignment between the theoretical setup and the algorithm's implementation regarding the proposal distribution for importance sampling. The loss function (Eq. 10) takes an expectation over samples drawn from $\pi_{sft}$, suggesting the importance weight $w(x, y_w, y_l)$ should correct for the ratio $\pi^*(y|x)/\pi_{sft}(y|x)$ (as derived in Appendix A.1). However, Algorithm 1 samples responses from the current policy $\pi_{\theta_t}$ (line 4). Applying weights derived assuming $\pi_{sft}$ to samples drawn from $\pi_{\theta_t}$ is incorrect and lacks clear justification. Furthermore, the notation $\mathbb{E}_ {(x,y_w,y_l)\sim \pi_{sft}}$ is imprecise, as $\pi_{sft}$ is a conditional distribution over $y$.
2. While positioned as an improvement for DPO-style methods (which are attractive for avoiding explicit reward models), ISO critically relies on an external, pre-trained reward model $r_\phi$ (Algorithm 1, line 5, Eq. 8) to compute the importance weights. This seems counter to the DPO philosophy and introduces a dependency not present in standard DPO. The paper does not clarify if this requires additional reward model training specific to the online setting. Using the DPO implicit reward $r(x,y) \propto \log (\pi_{\theta_t}(y|x)/\pi_{ref}(y|x))$ instead would likely be a poor proxy for the optimal reward $r^\star$ needed to estimate $\pi^*$, undermining the theoretical basis of the importance weights.
3. Importance sampling can suffer from high variance, particularly if the sampling distribution ($\pi_\theta$ or $\pi_{sft}$) is very different from the target distribution ($\pi^*$). While potentially better than RSO's rejection rate, the paper could discuss potential variance issues and how they are managed.
4. The signed margin score $(\sigma(r_w - r_l) - 0.5)$ is introduced somewhat heuristically to upweight informative pairs. While intuitive, a more formal justification or an ablation study isolating its specific impact on performance and variance would strengthen this component.

**Questions:**

Please see the weaknesses part above.

---

> ### Author Response · Authors · 2025-11-21
> **Response to Reviewer YRTE**
>
> We appreciate the reviewer’s sharp eye regarding the theoretical consistency and the constructive feedback on our experimental ablations. We address each point below.
>
> > There is a critical misalignment between the theoretical setup and the algorithm's implementation... Applying weights derived assuming $\pi_{sft}$ to samples drawn from $\pi_{\theta_t}$ is incorrect.
>
> We thank the reviewer for identifying this notational inconsistency. You are correct: the derivation in the Appendix describes the static case (sampling from $\pi_{sft}$), while Algorithm 1 describes the iterative case (sampling from the current policy $\pi_{\theta_t}$).
>
> In the final revision, we will generalize the derivation to the iterative setting. Specifically, we will denote the sampling distribution at step $t$ as $\pi_{\theta_t}$. The importance weight definition will be updated to reflect the correction for the distribution $\pi_{\theta_t}$. We will also correct the notation in Equation 10 to explicitly denote the expectation over $\pi_{\theta_t}$, ensuring the theoretical setup strictly aligns with the implementation in Algorithm 1.
>
>
> > ISO critically relies on an external, pre-trained reward model... This seems counter to the DPO philosophy.
>
> We respectfully clarify that the dependence on a reward model is a necessity of the **Online** problem setting, not a limitation specific to ISO.
>
> Unlike *Offline* DPO (which uses a static, pre-labeled dataset), **Online** Preference Learning requires labeling newly generated responses in real-time. Since human labeling is too slow for the training loop, **all** online methods (Online DPO, RSO, OAIF) must utilize a proxy reward model to provide feedback signals.
>
> Therefore, ISO does not introduce an *additional* dependency; it simply utilizes the existing proxy reward model (already required for generating preference labels in the baseline Online DPO) to compute importance weights. No extra training is required beyond the standard proxy reward model setup.
>
> > Importance sampling can suffer from high variance... the paper could discuss potential variance issues and how they are managed.
>
> We agree that variance is a key challenge in importance sampling. We manage this through two specific mechanisms and validate it empirically:
>
> 1.  **Self-Normalization:** As shown in Eq. 9 (and Eq. 19 in Appendix), we normalize weights within each prompt batch. This bounds the weights and significantly improves stability compared to unnormalized importance sampling.
> 2.  **Sample Size Scaling:** In **Table 2**, we explicitly analyze the impact of sample size on performance. The results show that increasing the sample size (e.g., from 8 to 16) effectively reduces the variance of the estimator, leading to higher win rates. We found that 16 samples provide a sweet spot where the variance is sufficiently low to achieve strong performance (outperforming RSO) without incurring excessive computational costs.
>
> > The signed margin score... is introduced somewhat heuristically... a more formal justification or an ablation study... would strengthen this component.
>
> We agree that isolating the impact of the signed margin score is important. We have conducted this ablation study and present the results in **Table 1** of our General Response.
>
> The results demonstrate that the signed margin score provides a consistent performance improvement. For example, on the Gemma2-2B model (Iteration 1), removing the signed margin score results in a win rate drop from **87.66%** to **83.36%**.
>
> Intuitively, this term acts as a variance reduction heuristic: pairs with small reward margins (near zero) are noisy and uninformative. By down-weighting them, the optimization focuses on pairs where the preference signal is distinct, stabilizing the learning process.

---

### Official Review · Reviewer_xEth · 2025-11-05

**Soundness:** 1
**Presentation:** 3
**Contribution:** 1
**Rating:** 2
**Confidence:** 4

**Summary:**

The authors propose to weight the SLiC loss by the magnitude of the reward scores.

**Strengths:**

N/A

**Weaknesses:**

I reviewed an earlier version of this paper where I raised several concerns. Unfortunately, it does not seem like any of them have been addressed. Thus, I am repeating them verbatim below:

(-) I really did try my best here but I don't think there's any reasonable interpretation of what the authors are doing as importance sampling. There is literally no ratio of two distribution's probabilities -- they never divide out the SFT policy's probabilities. I spent some time trying to do mental gymnastics to justify the product of factors that are used as weights as legitimate importance weights in any sense and I couldn't get that math to work out either. At best, I can say they weighted a usually unweighted loss.

(-) Off the top of my head, I think the most natural baseline here is https://arxiv.org/abs/2404.16767, which also essentially uses a weighted DPO-like loss. I would suggest including it in future experiments.

**Questions:**

(1) Is there a way to prove your re-weighting scheme is an unbiased estimate of importance weights $w(x, y) = \frac{\pi^{\star}(y|x)}{\pi_{sft}(y|x)}$?

---

> ### Author Response · Authors · 2025-11-21
> **Response to Reviewer xEth**
>
> > There is literally no ratio of two distribution's probabilities -- they never divide out the SFT policy's probabilities
>
> We respectfully point out that our method is indeed a rigorous application of importance sampling, and the "missing division" the reviewer notes is actually the result of an exact algebraic cancellation.
>
> The importance weight is defined as the ratio of the target distribution to the proposal distribution: $w(x,y) = \frac{\pi^\star(y|x)}{\pi_{sft}(y|x)}$.
>
> In the standard KL-regularized RLHF framework (e.g., DPO, formulation), the optimal target policy $\pi^\star$ is analytically defined as:
> $\pi^\star(y|x) = \frac{1}{Z(x)} \pi_{sft}(y|x) \exp\left(\frac{r(x,y)}{\beta}\right)$
>
> When we substitute this definition of $\pi^\star$ into the importance weight formula, the $\pi_{sft}$ term in the numerator cancels exactly with the $\pi_{sft}$ term in the denominator:
>
> $w(x,y) = \frac{\frac{1}{Z(x)} \pi_{sft}(y|x) \exp(\frac{r(x,y)}{\beta})}{\pi_{sft}(y|x)} = \frac{1}{Z(x)} \exp\left(\frac{r(x,y)}{\beta}\right)$
>
> Thus, we **do** divide out the SFT policy’s probabilities, but because the optimal policy is defined relative to the SFT policy, the resulting weight effectively depends only on the reward and the partition function. This derivation is detailed in **Appendix A.1.3 (Eq. 16)**.
>
>
> > I think the most natural baseline here is https://arxiv.org/abs/2404.16767
>
> We thank the reviewer for pointing out REBEL (arXiv:2404.16767) as a relevant baseline. We agree that comparing against other weighted DPO variants strengthens the paper.
>
> Per your suggestion, we implemented the REBEL loss and evaluated it on the Ultrafeedback dataset using the same online setup as our main experiments. The results (presented in **Table 1** of our General Response) show that ISO significantly outperforms REBEL across both Gemma-2-2B and Qwen-2.5-3B models.
>
> We hypothesize that while REBEL effectively handles noisy labels via regression, ISO provides a more direct mechanism for handling the specific **distributional shift** inherent in online sampling. By explicitly correcting the mismatch between the sampling policy $\pi_{\text{sft}}$ (or $\pi_{\theta}$) and the target optimal policy $\pi^\star$ via importance sampling, ISO allows the model to learn more efficiently from generated data than the regression-based weighting used in REBEL.
>
>
>
> > Is there a way to prove your re-weighting scheme is an unbiased estimate of importance weights
>
>
> To be precise, while Equation (15) in the Appendix establishes the theoretical unbiasedness of the importance sampling *identity* (assuming access to the exact partition function $Z(x)$), our practical implementation utilizes **Self-Normalized Importance Sampling (SNIS)** (Eq. 9 and Eq. 19) because the partition function $Z(x)$ for the optimal policy is intractable.
>
> Therefore, our estimator is technically **biased for finite sample sizes ($K$) but is statistically consistent**. As the number of samples $K \to \infty$, the estimator converges to the true expectation under $\pi^\star$. We chose SNIS because it is a standard approach in off-policy evaluation (e.g., standard implementations of weighted importance sampling) to handle unknown normalization constants.
>
> Furthermore, regarding the signed margin score (Eq. 8), we view this as a design choice to reduce the variance of the estimator by prioritizing informative pairs, effectively trading a small amount of bias for significantly better optimization stability and sample efficiency, as evidenced by our empirical results in Table 1 and Table 2.

---

### Official Review · Reviewer_zHkZ · 2025-11-06

**Soundness:** 2
**Presentation:** 2
**Contribution:** 1
**Rating:** 4
**Confidence:** 3

**Summary:**

The paper proposes a new variant of online DPO method where an importance ratio term is introduced such that the update is performed under the optimal policy's generation distribution. The paper performs experiments that the ISO outperforms online DPO or rejection sampling.

**Strengths:**

1. The paper makes an interesting observation that, even though one can not sample from the optimal policy, one can still evaluate the optimal policy's density, thus enabling the importance ratio correction.

2. According to the presented experiment results, ISO outperforms online DPO and iterative rejection sampling.

**Weaknesses:**

1. The method requires the access to the ground truth reward. With a reliable reward, one can simply perform online RL instead of contrastive learning.

2. It is unclear the benefit of the importance sampling as it increases the variance of the estimator.

3. The experiments are only performed for 2 iterations.

4. The presentation of the paper seems unpolished, for example, line 225 is unfinished, and eq 3 should not be $\mathcal{L}_{\mathrm{DPO}}$.

5. The paper is confusing the optimal policy and the optimal KL regularized policy. In the importance ratio correction, the optimal KL regularized policy is used.

**Questions:**

How important is the modulating factor? This ablation seems missing from the experiments.

---

> ### Author Response · Authors · 2025-11-21
> **Response to Reviewer zHkZ**
>
> We thank the reviewer for their detailed feedback and for identifying areas where our presentation and experimental scope could be improved. We address your specific concerns below.
>
> > The method requires the access to the ground truth reward. With a reliable reward, one can simply perform online RL instead of contrastive learning.
>
> We would like to clarify a misunderstanding regarding the reward setup. Our method does **not** require access to a ground truth (oracle) reward.
>
> Consistent with standard RLHF and Online DPO frameworks (e.g., Liu et al., 2023; Guo et al., 2024), we utilize a **learned proxy reward model** ($\rho_{\psi}$) trained on human preference data to derive the target policy and compute importance weights. This proxy model is distinct from the "golden reward model" used solely for final evaluation.
>
> Furthermore, while Online RL (like PPO) is a valid alternative, it is often computationally heavier and more unstable than contrastive/preference-based methods. ISO is designed to improve the sample efficiency of these preference-based methods (like DPO/RSO) by better utilizing the data generated by the current policy, without requiring the infrastructure complexity of PPO.
>
> > It is unclear the benefit of the importance sampling as it increases the variance of the estimator.
>
> This is a classic trade-off in off-policy learning: we accept increased variance to reduce **bias**.
>
> Standard online methods (like the "Direct" baseline) sample from $\pi_{\theta}$ but treat the data as if it came from the optimal policy $\pi^\star$, leading to a distribution mismatch (bias). ISO uses importance sampling to correct this bias, mathematically aligning the gradient updates with the target optimal distribution.
>
> To manage the variance inherent in importance sampling, we employ two specific techniques:
> 1.  **Self-Normalization (Eq. 9):** We normalize weights within each prompt batch, which improves stability.
> 2.  **Signed Margin Score (Eq. 8):** As detailed in our ablation studies, this term down-weights pairs with low reward margins (which often have high variance/noise), effectively acting as a variance reduction control.
>
> Empirically, our results (Table 1) show that ISO outperforms RSO and Direct methods, suggesting that the reduction in bias outweighs the impact of variance.
>
>
> > The experiments are only performed for 2 iterations.
>
> We restricted our experiments to 2 iterations to ensure a fair evaluation against the "golden" reward model. In proxy-guided optimization, it is a well-documented phenomenon (often called "reward hacking" or Goodhart's Law) that optimizing against a fixed proxy reward model for too many iterations eventually degrades true performance (golden reward), even as proxy reward scores continue to rise.
>
> Our results in Table 1 and Table 2 show that even within these 2 iterations, ISO learns significantly more efficiently than baselines. We will add a discussion on reward hacking to the limitations section to clarify this design choice.
>
> > The presentation of the paper seems unpolished, for example, line 225 is unfinished, and eq 3 should not be
>
> We apologize for the oversight on **Line 225** and will correct the unfinished sentence in the final revision.
>
> Regarding **Equation 3**: This equation represents the **hinge-loss variant** of the DPO objective, similar to the SLiC loss (Zhao et al., 2023), which we included for completeness as ISO is compatible with various loss functions. We will review the notation to ensure it aligns strictly with standard conventions and improves clarity.
>
> > The paper is confusing the optimal policy and the optimal KL regularized policy. In the importance ratio correction, the optimal KL regularized policy is used.
>
> You are technically correct that $\pi^\star$ in our derivation refers to the optimal KL-regularized policy. In Appendix A.1.1 (Eq. 11), we explicitly define $\pi^\star(y|x) \propto \pi_{sft}(y|x) \exp(r(x,y)/\beta)$. This is the closed-form solution to the standard KL-constrained reward maximization problem (as derived in DPO). When we refer to the "optimal policy" throughout the paper, we are referring to this specific KL-regularized distribution. We will update the terminology in Section 3.1 and the Appendix to be precise that $\pi^\star$ refers to the "optimal KL-regularized policy."
>
> > How important is the modulating factor? This ablation seems missing from the experiments.
>
> We have performed this ablation and included the results in **Table 1 of the General Response**.
>
> The results show that the modulating factor (the Signed Margin Score) is indeed a critical component. For example, on the Gemma2-2B model (Iteration 1), removing the signed margin score drops the win rate from **87.66% (Full ISO)** to **83.36% (ISO w/o Margin)**. This confirms that down-weighting low-confidence pairs helps stabilize the importance sampling process.

---

### Author Response · Authors · 2025-11-21
**General Response**

## Comparison with Weighted DPO Baselines and Ablation Studies


To address reviewer questions regarding how ISO compares to other weighted loss formulations (specifically **REBEL**; arXiv:2404.16767) and to further validate the components of our method, we conducted additional experiments on the Ultrafeedback dataset.

We compare the following settings:
1.  **Direct:** Standard DPO with uniform weighting on online samples.
2.  **REBEL:** The regression-based weighted DPO loss.
3.  **ISO w/o Signed Margin:** Our importance sampling method using only the base weights derived from the partition-function-free estimator (Eq. 19), without the signed margin modulation.
4.  **ISO (Ours):** The full method including the signed margin score (Eq. 8).

We set the $\gamma=0.5$ in the ISO experiment.

**Table 1: Proxy Reward Win Rates (%) on Ultrafeedback**

| Iteration | Model | Direct | REBEL | ISO w/o Signed Margin | ISO (Ours) |
| :--- | :--- | :--- | :--- | :--- | :--- |
| **1** | **Gemma2 2B** | 80.77 | 71.87 | 83.36 | **87.66** |
| | **Qwen2.5 3B** | 79.74 | 69.38 | 79.03 | **82.10** |
| **2** | **Gemma2 2B** | 89.47 | 82.43 | 91.38 | **94.34** |
| | **Qwen2.5 3B** | 90.65 | 81.40 | 87.83 | **90.75** |

**Analysis of Results:**
* **ISO vs. REBEL:** ISO consistently outperforms REBEL in all settings (e.g., **+15.79%** for Gemma2 2B in Iteration 1). While REBEL focuses on regressing the reward margin, our results suggest that ISO's importance sampling approach is more effective at bridging the gap between the online exploration policy and the optimal target policy.
* **Impact of Signed Margin Score:** Comparing "ISO w/o Signed Margin" to the full "ISO," we observe that the signed margin score contributes consistently to performance (e.g., increasing win rate from 83.36% to 87.66% on Gemma2 2B). This confirms that down-weighting pairs with low reward margins helps the model focus on more informative preference pairs.
* **Importance Sampling Baseline:** Even without the signed margin score, the core importance sampling mechanism ("ISO w/o Signed Margin") generally performs on par with or better than the Direct baseline, validating the effectiveness of our derived importance weights in correcting distribution mismatch.

---

### Meta-Review · Area_Chair_SNVE · 2026-01-01

**Summary:**

The major concerns from reviewers are mainly about novelty and clarification. For the novelty, two reviewers point out that the contribution is limited because the importance sampling setting is straightforward. The clarification is a very negative issue. Some clarification of the method and experiments makes it hard to distinguish the contribution.

**Reviewer Concerns:**

Some clarifications on the proposed method will partially address the authors' concerns, while the contribution remains a remaining issue, which is difficult to address in this version.

**Reviewer Scores:**

Reviewers zHkZ, xEth, and J7oq express their concerns about the contribution, which is hard to address, and their scores will remain.

Reviewer YRTE may raise his score due to some concerns are partially addressed.

---

### Decision · Program_Chairs · 2026-01-26

Reject